# Assessing the Effectiveness of Fermented Banana Peel Extracts for the Biosorption and Removal of Cadmium to Mitigate Inflammation and Oxidative Stress

**DOI:** 10.3390/foods12132632

**Published:** 2023-07-07

**Authors:** Lan-Chun Chou, Cheng-Chih Tsai

**Affiliations:** Department of Food Science and Technology, HungKuang University, Shalu District, Taichung City 43302, Taiwan

**Keywords:** lactic acid bacteria, cadmium, banana peel, oxidative stress, inflammatory, Fourier transform infrared spectroscopy

## Abstract

This study identified 11 lactic acid bacteria (LAB) strains that exhibited tolerance to heavy metal cadmium concentrations above 50 ppm for 48 h. Among these strains, T126-1 and T40-1 displayed the highest tolerance, enduring cadmium concentrations up to 500 ppm while still inhibiting bacterial growth by 50%. Moreover, the fermentation of banana peel using LAB significantly enhanced the clearance rate of cadmium (*p* < 0.05) compared to nonfermented banana peel. Additionally, the LAB-fermented banana peel exhibited higher 1,1-diphenyl-2-picryl-hydrazyl (DPPH) and reduced power values. Strain T40-1 exhibited a significant improvement in its ability to chelate ferrous ions (*p* < 0.05). Regarding antibiotic resistance, both the T40-1 and TH3 strains demonstrated high resistance with a third-level inhibition rate against ampicillin and tetracycline. Cell viability tests revealed that incubation with the T40-1 and TH3 strains for a duration of 24 h did not result in any cellular damage. Moreover, these LAB strains effectively mitigated oxidative stress markers, such as reactive oxygen species (ROS), glutathione (GSH), and lactate dehydrogenase (LDH), caused by 2 ppm cadmium on cells. Furthermore, the LAB strains were able to reduce the inflammatory response, as evidenced by a decrease in interleukin-8 (IL-8) levels (*p* < 0.05). The use of Fourier transform infrared (FT-IR) spectroscopy analysis provided valuable insight into the interaction between metal ions and the organic functional groups present on the cell wall of fermented banana peel. In summary, this study highlights the potential of the LAB strains T40-1 and TH3 in terms of their tolerance to the cadmium, ability to enhance cadmium clearance rates, and their beneficial effects on oxidative stress, inflammation, and cell viability.

## 1. Introduction

Cadmium compounds are prevalent in nature and have found extensive use in various industries because of their corrosion resistance properties [1]. Common applications of cadmium include nickel–cadmium batteries, plastic stabilizers, dyes, metal alloys, and electroplating [1]. However, the discharge of cadmium and its compounds from industrial sources and waste presents a significant environmental concern. These pollutants often contaminate natural water bodies and soils, leading to potential ecological and health risks. The stable chemical nature of cadmium enables its accumulation as residues, which can be absorbed by crops and subsequently transferred through the food chain, resulting in bioaccumulation issues [2]. Numerous studies have shed light on the global prevalence of cadmium pollution, particularly in areas where factory wastewater contaminates irrigation water and farmland in the vicinity [3]. The consumption of crops and drinking water contaminated with cadmium poses risks to human health, potentially causing cadmium poisoning, bone lesions, severe pain, and the development of itai-itai disease—a condition associated with chronic cadmium exposure [4]. It is crucial to address and mitigate the release of cadmium into the environment to safeguard both ecosystems and human well-being.

Cadmium, a nonessential metal, poses significant toxicity risks to the human body. Exposure to cadmium can occur through various routes, including inhalation, skin contact, and gastrointestinal absorption [5]. Upon entering the body, cadmium tends to accumulate primarily in the liver and kidneys. Although excretion through feces and urine is possible, the efficiency is remarkably low, resulting in a half-life of cadmium ranging from 10 to 30 years [6]. The adverse effects of cadmium exposure on human health are substantial and can even contribute to the development of cancer [7]. Research suggests that cadmium toxicity primarily occurs through the generation of reactive oxygen species (ROS), leading to increased oxidative stress, depletion of reduced glutathione (GSH), and disruption of the nuclear factor erythroid-2-related factor (Nrf2) binding to the antioxidant response element (ARE). Consequently, the expression of downstream antioxidant-related genes is affected, impairing the cellular antioxidant defense system [8,9].

Detoxification mechanisms serve as protective measures to prevent the entry of harmful compounds into the body [10]. Among these mechanisms, the metal-binding capability of intestinal flora, including probiotics such as *Lactobacillus*, plays a crucial role in blocking the absorption of toxic substances and safeguarding the host. Probiotics, known for their generally recognized safe (GRAS) status in the food industry, exhibit metal-binding and chelation properties, effectively reducing the uptake of heavy metal ions and toxins into the human body. Additionally, microorganisms possess intricate mechanisms to regulate metal ion concentrations, sense internal and external metal levels, activate relevant genes, and produce channel proteins with varying affinity sites [11].

Studies have demonstrated the ability of probiotics, such as *Lactobacillus plantarum*, *L. rhamnosus*, *Bifidobacterium breve*, and *B. lactis*, to remove heavy metals through biosorption in vitro [12,13]. Apart from their inherent metal-adsorbing capabilities, the exopolysaccharides (EPS) secreted by microorganisms during growth have been reported to possess biosorbent properties for heavy metals [14]. The interaction between metal cations and the negatively charged acidic functional groups of EPS accounts for their heavy metal adsorption capacity. EPS contains acidic functional groups, such as carbonyl, ester, hydroxylamine, phosphate, and sulfate groups, which serve as potential binding sites for heavy metals [15]. Numerous scientific investigations have highlighted the diverse potential applications of banana peel, ranging from its use as a food additive in healthcare and medicine to its efficacy as a heavy metal adsorbent, biofuel source, and dietary fiber extraction [16]. Particularly, banana peels contain approximately 10–21% pectin, a semi-acidic polysaccharide composed of lacturonic acid, galactose, and rhamnose [17,18]. The presence of galacturonic acid with carboxyl groups [19] and amine groups in the protein components of banana peels [20] contributes to their effective adsorption of multivalent metal cations in aqueous solutions. Furthermore, the cellulose components of banana peel, such as lignin and hemicellulose, have demonstrated the ability to adsorb heavy metals.

The aim of this study was to evaluate the adsorption capacity and elimination potential of banana peel fermented with LAB against the cadmium while also assessing their ability to mitigate inflammation and oxidative damage. The screening process involved the selection of LAB strains that exhibited tolerance to cadmium and possessed desirable adsorption characteristics. Additionally, the chosen LAB strains should demonstrate traits such as resistance to acidic and bile salt environments, intestinal adhesion, safety, and antibiotic resistance. The optimal fermentation conditions for banana peels and functional LAB strains were determined, and the production of the fermented product was simulated. Using Fourier transform infrared spectroscopy (FT-IR), the functional group changes in the banana peel fermentation products before and after cadmium adsorption were determined. The ability of the fermented product to scavenge free radicals and inhibit lipid peroxidation was also analyzed. The findings will contribute to the development of sustainable and eco-friendly approaches for heavy metal remediation and human health protection.

## 2. Materials and Methods

### 2.1. Strain Cultivation

LAB strains isolated from pickles, fermented products, and infant feces were utilized in this study. The strains were cultured by activation in Lactobacilli MRS broth (DIFCO, Detroit, MI, USA) under the conditions of 37 °C for 20 h. Subsequently, the strains were preserved in MRS broth supplemented with 25% glycerol and stored at −80 °C.

### 2.2. Screening for LAB Strains with Cadmium Tolerance

The cadmium tolerance of the LAB strains was determined by assessing the minimum inhibitory concentration (MIC_50_), which inhibits 50% of the strain growth [21]. Cadmium chloride solution was sterilized using a sterile filter membrane with a pore size of 0.22 μm and added to a 96-well microplate containing 10^5^ CFU/mL LAB. The initial concentration was set at 1000 mg/L, and subsequent serial dilutions were prepared, halving the concentration to achieve a range from 6.25 mg/L. The growth of the LAB was monitored after 24 and 48 h of incubation at 37 °C. The MIC_50_ was determined as the lowest concentration of cadmium metal that inhibited 50% of the strain growth.

### 2.3. Screening for LAB Strains with Cadmium Adsorption Potential

After 18 h of LAB cultivation, the bacterial cells were collected by centrifugation and washed twice with ultrapure water. The cells were then resuspended in ultrapure water containing 50 mg/L cadmium chloride. The pH of the bacterial suspension was adjusted to 6 using NaOH or HNO_3_, followed by incubation at 37 °C for 3 and 24 h. After the incubation period, the supernatant was collected by centrifugation, and the concentration of residual cadmium metal was measured using flame atomic absorption spectrophotometry (Spectr AA 220; Varian) (Scientific Equipment Ltd., Dandenong, Australia). The cadmium metal adsorption capacity of the strain was calculated as the percentage of cadmium metal removal using the following formula: Removal (%) = 100% ((C_0_ − C_1_)/C_0_), where C_0_ and C_1_ represent the initial and residual cadmium metal concentrations, respectively.

### 2.4. Acid Resistance, Bile Salt Resistance, and Antibiotic Tolerance Tests

To evaluate the acid resistance, phosphate buffer with pepsin at pH 3.0 was added to 1000 µL of LAB suspension (10^8^ CFU/mL), and the bacteria were incubated at 37 °C for 0, 1.5, and 3 h. As a control, the LAB suspension mixed with phosphate buffer at pH 7.0 was cultured under the same conditions. After incubation, the viable LAB counts were determined using the pour plate method on MRS agar plates. For the bile salt resistance test, 1 mL of LAB suspension treated with acid for 3 h was added to 9 mL of phosphate buffer with or without 0.3% ox gall bile (Sigma-Aldrich Corp., St. Louis, MO, USA) and 0.1% pancreatin. The mixture was cultured for 0, 1.5, and 3 h, followed by an enumeration of the LAB counts using the pour plate method on MRS agar plates [22].

Antibiotic tolerance was assessed by immersing sterile cotton swabs in the cultured LAB suspension (10^7^ CFU/mL) and streaking them evenly over the agar surface in Petri dishes. Antibiotic paper discs (ingots) were placed on the agar surface using sterilized tweezers, ensuring full contact. The Petri dishes were incubated upside down overnight, and the diameter of the inhibition zone was measured and recorded. The following antibiotics and doses were used: ampicillin (10 μg), kanamycin (30 μg), tetracycline (30 μg), penicillin G (10 units), neomycin (30 μg), erythromycin (15 μg), streptomycin (10 μg), and gentamicin (30 μg).

### 2.5. Production of Fermented LAB Powder from Banana Peel

After determining the optimal growth medium for the LAB fermentation with banana peel, the fermentation broth was collected after 16 h of cultivation to assess the pH, OD value, and LAB count. Once the cultivation was confirmed to be complete, the fermentation broth was subjected to freeze-drying. The resulting freeze-dried product was crushed and sieved to obtain the fermented LAB powder derived from banana peel.

### 2.6. Antioxidant Activity Assay

#### 2.6.1. 2,2-Diphenyl-1-picrylhydrazine (DPPH) Scavenging Assay

A mixture of 1 mL of the sample and 1 mL of freshly prepared DPPH solution (0.2 mM methanol solution; Sigma, USA) was incubated in the dark for 30 min. A mixture of DPPH and PBS (pH 7.2) served as the blank sample. After centrifugation at 7000 rpm for 10 min, the DPPH content was quantified by measuring the absorbance at a specific wavelength of 517 nm. The scavenging capacity was calculated as follows: clearance (%) = (1 − A_517_(sample)/A_517_(blank)) × 100.

#### 2.6.2. Hydroxyl Radical Scavenging Assay

A mixture of 1 mL of 1,10-phenanthroline, 1 mL of PBS (pH 7.2), 1 mL of the sample, and 1 mL of FeSO_4_ was prepared. Subsequently, 1 mL of H_2_O_2_ was added, and the mixture was incubated at 37 °C for 1.5 h. The absorbance of the mixture was measured at a specific wavelength of 536 nm. A volume of distilled water equal to the sample volume was used as the blank group. The clearance capacity was calculated as follows: clearance rate (%) = (A_536_(sample)/A_536_(blank))/(A_536_(control)/A_536_(blank)) × 100.

#### 2.6.3. Fe^2+^ Chelating Activity

The sample was reacted with 2 mM ferrous chloride for 30 s, followed by the addition of 5 mM ferrozine. The mixture was incubated at room temperature for 10 min, and the absorbance was immediately measured at 562 nm using a spectrophotometer. A higher absorbance reading indicates a stronger ability to chelate ferrous ions.

#### 2.6.4. Total Reducing Power

The LAB fermentation product was mixed with 0.2 M phosphate buffer (pH 6.6) and 1% red blood salt. The mixture was then cooled in a 50 °C water bath for 20 min. Subsequently, 10% trichloroacetic acid (TCA) solution was added, followed by centrifugation at 2000× *g* for 20 min. The supernatant was collected and mixed with 0.1% ferric chloride (FeCl_3_) solution. The absorbance of the mixture was measured at 700 nm using a spectrophotometer, with a higher absorbance indicating stronger reducing power.

### 2.7. Analysis of Caco-2 Cell Survival Rate Using Fermented Banana Peel LAB Powder

One hundred μL (3 × 10^4^ cells/well) of human intestinal epithelial Caco-2 cells were seeded into 24-well plates. After complete differentiation, the culture medium was replaced with fresh medium without fetal bovine serum, and different treatment samples were added for a 24 h incubation. For the analysis, the supernatant was aspirated, and the cells were washed twice with a phosphate-buffered solution (PBS). Then, 100 μL of 5 mg/mL MTT solution was added to each well and incubated at 37 °C for 30 min. After removing the supernatant, the formazan crystals were dissolved in 0.2 mL of dimethyl sulfoxide (DMSO) per well, and the absorbance was measured at a wavelength of 570 nm using an enzyme immunoassay reader [23].

### 2.8. Banana Peel LAB Fermentation Powder Inhibits Cadmium-Induced Inflammation in Caco-2 Cells

A total of 500 μL of banana peel fermented powder with or without LAB and 2 ppm cadmium were separately added to Caco-2 cells cultured in 24-well plates. Each well was supplemented with 1 mL of cell suspension and incubated at 37 °C in a 5% CO_2_ incubator until a monolayer was formed. Cells were pretreated with or without lactic acid bacteria to evaluate the reduction in cadmium-induced damage. After 24 h of coculture, the cell culture medium was collected and stored at −80 °C. The pro-inflammatory cytokine IL-8 was analyzed using a commercially available ELISA kit [24].

### 2.9. Analysis of Cell Oxidative Damage

#### 2.9.1. Intracellular Oxidative Damage Assay

Human intestinal epithelial cells were cultured in DMEM containing 10% FBS and adjusted to a concentration of 3 × 10^5^ cells/mL. The cells were seeded into 6 cm culture dishes and incubated in a 37 °C, 5% CO_2_ incubator until complete differentiation. Subsequently, the cells were treated with or without lactic acid bacteria-fermented banana peel products and 2 ppm cadmium to evaluate their effect on reducing cellular damage caused by cadmium. After the incubation period, the ROS detection kit was added to assess intracellular reactive oxygen species levels.

In the MDA determination step, the cells were treated with cadmium and the sample. Following treatment, the cells were scraped off and mixed with 100 μL of 50% TCA (final concentration 0.5%). The mixture was shaken, followed by centrifugation for 20 min at 1500× *g*. To 1 mL of the supernatant, 1.1 mL of 0.6% TBA was added, and the mixture was heated in a water bath at 100 °C for 1 h. After cooling on ice, 3 mL of isobutanol was added and shaken for 2 min. The mixture was then centrifuged for 20 min at 5000 rpm, and the cell extracts were collected for MDA content determination [25].

For the LDH assay, the cells were cultured in a 96-well plate, and the LAB samples were added. The supernatant was collected, and 50 μL was transferred to a new 96-well plate. An additional well containing 10× lysis buffer was prepared as a positive control. In a light-proof environment, 50 μL of substrate mix was added to each well and incubated at 37 °C for 20 min. Subsequently, 50 μL of stop solution was added, and the absorbance was measured at 490 nm. The LDH cytotoxicity was calculated using the formula: cytotoxicity (%) = (experimental group/control group) × 100.

#### 2.9.2. Intracellular Antioxidant Activity Assay

Human intestinal epithelial cells were cultured and treated as described above. After incubation, the commercially available GSH detection kit’s fluorescent reagent was added, and the cell lysates were collected to measure the intracellular GSH activity.

### 2.10. FT-IR Analysis

The sorption process of fresh-dried and metal-laden fermented banana peel was analyzed using FT-IR to identify the functional groups involved. A Fourier transform infrared spectrometer, FT/IR-460 (Jusco, Tokyo, Japan), was used for this analysis. KBr disks were prepared by combining 200 mg of each sample with 1% ground powder.

### 2.11. Statistical Analysis

The data obtained in this study were analyzed using the IBM Statistical Package for the Social Sciences (SPSS) Statistics for Windows, Version 27.0. Armonk, NY, USA: IBM Corp. The experimental results are expressed as the mean ± standard deviation (SD). Duncan’s multiple range test was conducted to compare the mean values of each experimental group, and a significance level of *p* < 0.05 was used as the criterion for determining significant differences.

## 3. Results and Discussion

### 3.1. Screening of LAB with Cadmium Tolerance

The tolerance of LAB to cadmium was assessed, and the results are presented in Table 1. Eleven strains of LAB exhibited tolerance to cadmium concentrations above 50 ppm for a duration of 48 h. Strains T126-1 and T40-1 displayed the highest tolerance, with the ability to withstand cadmium concentrations up to 500 ppm while inhibiting bacterial growth by 50%. Other strains, including TH3, T16-1, B22, and TF-1, exhibited high tolerance and were able to withstand concentrations of 250 ppm of cadmium. Jaafar (2020) evaluated the tolerance of *L. acidophilus* to Cd by determining the minimum inhibitory concentration (MIC). The results indicated that the MIC value for Cd was 150 ppm. In this case, it can be inferred that the bacteria exhibited resistance to Cd [13].

### 3.2. Screening of LAB with Cadmium Adsorption Potential

Figure 1 illustrates the capacity of lactic acid bacteria to remove cadmium. When exposed to a cadmium chloride solution of 50 mg/L for three hours, the removal rate exceeded 60%, which further increased to 80% after continuous exposure for 24 h. Notably, strain T16-1 exhibited the most effective cadmium removal capacity (*p* < 0.05), achieving a removal rate of 96% after a three-hour incubation with cadmium.

Previous studies have highlighted the ability of specific strains, such as *Enterococcus faecium* EF031 and *E. faecium* M74, to bind heavy metals rapidly, forming complexes that can persist for at least 48 h and be excreted with the strain in feces [26,27]. The capacity of EPS to remove heavy metals has been extensively studied in various microorganisms under different conditions. For instance, the EPS of *Paenibacillus jamilae* can adsorb lead, cadmium, cobalt, nickel, zinc, and copper [28]; the EPS of *Aspergillus fumigatus* can adsorb copper(II) and cadmium(II) [29]; the EPS of *Lyngbya putealis* can adsorb chromium [30]; and LAB’s EPS is currently under investigation for its capacity to adsorb lead(II), aluminum, and cadmium [31].

### 3.3. Adsorption of Cadmium by LAB-Fermented Banana Peel

The banana peel was incubated with a 50 mg/L cadmium chloride solution for 3 and 24 h, resulting in a cadmium removal rate of approximately 65%. Upon addition of two LAB strains, TH3 and T40-1, to the fermented banana peel, the removal rates increased by 28% and 25%, respectively (Figure 2). These findings indicate that LAB-fermented banana peel significantly enhances the cadmium clearance rate (*p* < 0.05), and this effect remains robust even after prolonged exposure.

In prior research, 0.1 g of dried banana peel powder was added to a solution containing 50 ppm of Cd(II). The mixture was shaken at room temperature for half an hour, and the adsorption capacity reached 95% [16]. The ratio of banana peel to water in this experiment was 1:3. Microbial fermentation has been demonstrated to be a promising method for removing Cd from food [11]. Furthermore, studies have shown that microbial and plant inoculation can enhance the absorption of heavy metals in wastewater. The presence of these bacteria would potentially enhance the conversion of the more soluble form of chromate Cr (VI) to a less toxic and less mobile form of Cr (III) [21].

### 3.4. Determination of Antioxidant Activity in LAB-Fermented Banana Peel Products

Although the fermentation liquid of banana peel with two groups of LAB exhibited higher values of DPPH and reducing power compared to nonfermented banana peel juice, the difference was not statistically significant (*p* > 0.05). This suggests that banana peel itself possesses inherent antioxidant capacity. Strain T40-1 significantly enhanced the ability to chelate ferrous ions (*p* < 0.05), while strain TH3 significantly improved the ability to scavenge hydroxyl radicals (*p* < 0.05) (Table 2).

Cadmium poisoning is associated with oxidative stress, which can inhibit the production of antioxidant enzymes and reactive oxygen species (ROS) in the body, leading to lipid peroxidation, DNA oxidative damage, and organ tissue damage [32]. Previous studies have shown that a synbiotic diet consisting of the compound probiotics *L. plantarum* and *Bacillus coagulans* combined with the prebiotic inulin enhances antioxidant capacity, improves serum biochemical values, and reduces cadmium content in the liver and kidney when orally administered to Wistar rats exposed to cadmium [33]. The anionic groups on the bacterial surface, as well as peptidoglycan and teichoic acid in the cell wall, effectively bind heavy metals, facilitate their excretion from the body, and prevent damage while restoring the activity of antioxidant enzymes.

### 3.5. Determination of Probiotic Properties

To ensure efficacy, probiotics must survive the acidic environment of the stomach. In a simulated gastric solution with pH 3.0, two strains of lactic acid bacteria maintained a concentration of 8 log CFU/mL after 3 h. Additionally, the addition of a phosphate solution containing 0.3% bile salts did not reduce the bacterial count, indicating that the two strains were tolerant to bile salts (Table 3). The antibiotic paper tablet susceptibility test (Table 4) revealed that both the T40-1 and TH3 strains exhibited high resistance to kanamycin, streptomycin, gentamicin, neomycin, and erythromycin, with only a first-level inhibition rate. T40-1 and TH3 demonstrated a second-level inhibition rate for penicillin G and a third-level inhibition rate for ampicillin and tetracycline.

A high survival rate is an essential criterion for probiotics to be beneficial to the host during the passage through the gastrointestinal tract. Therefore, it is imperative that a potential probiotic strain is well tolerated in extreme environments in the gastrointestinal tract, such as low-pH gastric juices, high concentrations of bile salts, and prolonged periods of intestinal juice. Acid-tolerant strains have an advantage in surviving in the low pH conditions of the stomach, where hydrochloric and gastric acids are secreted. In general, the physiological concentration of human bile ranges from 0.3% to 0.5%. Therefore, resistance to bile acid is an important characteristic that enables probiotics to survive, grow, and remain active in the small intestine. Although such studies have major differences in species in design, they all show that acid and bile acid have separate and combined effects on bacterial growth. Therefore, the acid and bile tolerance demonstrated by the LAB studies here suggests that these strains are likely resistant to stomach and intestinal conditions [12,13].

### 3.6. Analysis of Caco-2 Cell Survival Rate Using LAB-Fermented Banana Peel Products

Based on previous experiments investigating the adsorption of cadmium and the antioxidant properties of LAB-fermented banana peel products, strain T40 was identified as the most effective. Subsequently, a cell test was conducted to evaluate the optimal concentration of the fermentation broth that would not cause cell damage (cell survival rate ≥ 90%). The results, as shown in Figure 3, indicated that both the banana peel fermented with LAB strain T40 and the banana peel fermented without strains after heat treatment exhibited cell tolerance up to 500 μL/mL. Therefore, this concentration was used for subsequent experiments.

### 3.7. Inhibition of Cadmium-Induced Inflammation in Caco-2 Cells by LAB-Fermented Banana Peel Product

Based on the experimental results presented in Figure 4, the addition of cadmium to cells for 24 h significantly induced the production of IL-8 cytokines (*p* < 0.05). However, both banana peel juice and banana peel fermentation broth (inactivated bacteria) effectively reduced the inflammation reaction induced by cadmium exposure. Similar findings were reported by Hyun et al., who observed that CdCl_2_ stimulated the secretion of IL-8 in Caco-2 cells, while the secretion of other inflammatory cytokines, such as TNF-α, IL-1β, and IFN-γ, remained unaffected by CdCl_2_ exposure (Figure 5).

IL-8 is a chemotactic cytokine that attracts and activates leukocytes at sites of acute inflammation. However, excessive production of IL-8 can have detrimental effects on inflamed tissues. The CdCl_2_-stimulated over-secretion of IL-8 may indicate intestinal epithelial injury and inflammatory responses, suggesting a significant alteration in the intestinal immune response [34]. Furthermore, banana peels contain various antioxidant substances, such as phenolic acids, catechols, tocopherols, phytosterols, and polysaccharides, which can scavenge free radicals, reduce inflammation and oxidative stress, and enhance the capacity of antioxidant defense enzymes [35,36].

### 3.8. Analysis of Oxidative Cell Damage Induced by Cadmium Using LAB-Fermented Banana Peel Products

The results of the experiment are presented in Figure 6. The addition of samples alone did not induce LDH production by cells. However, the addition of cadmium for 24 h significantly increased the concentration of LDH in the cell fluid (*p* < 0.05). LAB-fermented banana peel (inactivated bacteria) effectively reduced LDH production (*p* < 0.05). In the case of MDA, the concentration of cadmium used in this test may be low, resulting in no significant stimulating effect (Figure 7). The addition of cadmium to cells for 24 h induced the production of ROS. However, both banana peel juice and banana peel fermentation liquid effectively reduced oxidative damage caused by cadmium (Figure 8).

Cadmium increases the expression of the XDH (xanthine dehydrogenase) gene, leading to the conversion of xanthine dehydrogenase to xanthine oxidase, which generates superoxide free radicals that can increase oxidative stress in cells [37]. The abnormal increase in reactive oxygen species enhances oxidative stress, which can trigger lipid peroxidation, and oxidative damage to proteins and DNA, ultimately resulting in tissue damage or apoptosis [38,39].

### 3.9. Determination of Intracellular Antioxidant Activity of LAB-Fermented Banana Peel Products

Treatment with a low concentration of cadmium (2 ppm) for 24 h significantly increased the concentration of GSH in cells (*p* < 0.05). However, banana peel juice fermented with lactic acid bacteria (inactivated bacteria) effectively reduced GSH synthesis (*p* < 0.05), with the most pronounced effect observed in banana peel fermented liquid (inactivated bacteria) containing LAB (Figure 9). Cadmium toxicity is associated with ROS generation, leading to oxidative stress and GSH depletion. However, chronic exposure to low doses of cadmium can induce cadmium tolerance and toxin accumulation, triggering GSH synthesis followed by its consumption, resulting in increased oxidative stress [40]. Probiotics with antioxidant properties, such as *L. plantarum*, have been found to be effective in heavy metal poisoning by producing various metabolites with antioxidant activity, including GSH and folic acid [41]. GSH not only enhances the host’s antioxidant capacity to resist oxidative stress but also reduces the toxicity of heavy metals by displacing heavy metal ions, thus facilitating their removal [42].

The protective effect of *L. plantarum* CCFM8610 against cadmium metal exposure in HT-29 cells and male C57black/6 mice was studied. It was observed that LAB could alleviate cytotoxicity, reduce oxidative stress and inflammation, preserve connexin integrity, decrease intestinal permeability in epithelial cells and animals, promote cadmium excretion and fecal reduction, and inhibit intestinal cadmium absorption. LAB can also chelate cadmium in the intestine, enhancing antioxidant capacity and protecting the intestine from damage. Probiotics with strong cadmium binding and antioxidant capacity can serve as supplements for the prevention of oral cadmium poisoning [43]. Additionally, the protective effects of *L. plantarum* CCFM8610 on the intestinal flora and physiology of tilapia (*Oreochromis niloticus*) exposed to cadmium metals in water sources have been investigated. The results revealed that the consumption of LAB reduced fish mortality and the formation of intestinal pathogens, lowered cadmium levels in blood and tissues, and alleviated oxidative tissue stress. These findings have implications for the prevention of aquaculture diseases and food safety [44].

### 3.10. FT-IR Analysis

Fourier transform infrared (FT-IR) spectroscopy was employed to analyze the binding sites present on the cell surface and their involvement in the biosorption process [45]. The FT-IR transmission spectra of fermented banana peel exhibited various peaks corresponding to different functional groups and bonds, as depicted in Figure 10. Upon interaction with Cd ions, the peaks exhibited broadening and shifting. The region between 3700 and 3300 cm^−1^ represented the O-H and N-H stretching vibrations [46]. Prior to the adsorption process, the sorbent spectra showed broad and intense peaks within the wave numbers of 3300–3400 cm^−1^. After treatment with Cd, the same stretching vibration at that wavenumber persisted, while a new N-H bending vibration band emerged at 1571 cm^−1^. These spectral changes indicated that binding with Cd altered the hybridization type around nitrogen, weakening the N-H bond strength and transforming it into a bonded NH group. Consequently, a downshift in both the stretching and bending absorption bands occurred [47].

The band observed at 2923–2939 cm^−1^ indicated the symmetric or asymmetric stretching vibration of aliphatic acids [48]. The absorption band at 2925 cm^–1^ corresponded to the C-H stretching vibrations of methyl or methylene groups commonly present in hexoses (such as glucose or galactose) and deoxyhexoses (such as rhamnose or fucose) [49]. In the range of 1800–1500 cm^−1^, characteristic bands for proteins were observed, with sharp peaks appearing in the cadmium treatment group between 1725 and 1600 cm^−1^, which is specific to amide-I bands. This region (1700 to 1600 cm^−1^) is known for C=O stretching vibrations of the peptide bonds [50]. Furthermore, the range from 1600 to 1500 cm^−1^ corresponds to amide-II bands, attributed to the N-H bending vibrations [51].

The stretching between 1366 and 1384 cm^−1^ indicates the bending vibrations of S=O in the sulfonic acid and sulfonate groups, which were found to be associated with the sorption of four metals (Cu^2+^, Cd^2+^, Zn^2+^, and Pd^2+^) [52]. The peak observed at 1058–1061 cm^−1^ was assigned to the asymmetric stretching vibration of C-OH in the alcoholic groups, and it may be attributed to carboxylic acids [48]. These spectral changes can be attributed to the interaction between metals and the organic functional groups present on the cell wall.

## 4. Conclusions

The findings suggest that lactic acid bacteria strains have the potential to be used for cadmium removal, as well as for improving the antioxidant and probiotic properties of fermented banana peel products.

## Figures and Tables

**Figure 1 foods-12-02632-f001:**
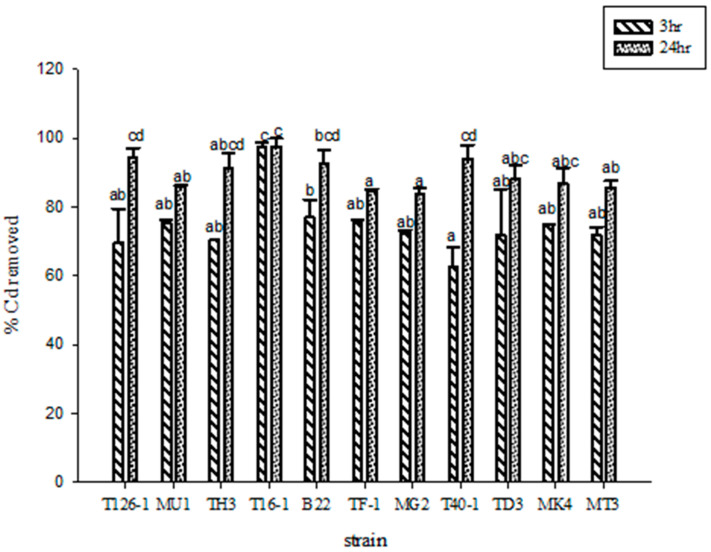
Lactic acid bacteria strains with 50 mg/L cadmium chloride solution were added for 3 h and 24 h after the scavenging rate of cadmium. ^a,b,c,d^ Values with different letters indicate a significant difference (*p* < 0.05) using Duncan’s multiple range test.

**Figure 2 foods-12-02632-f002:**
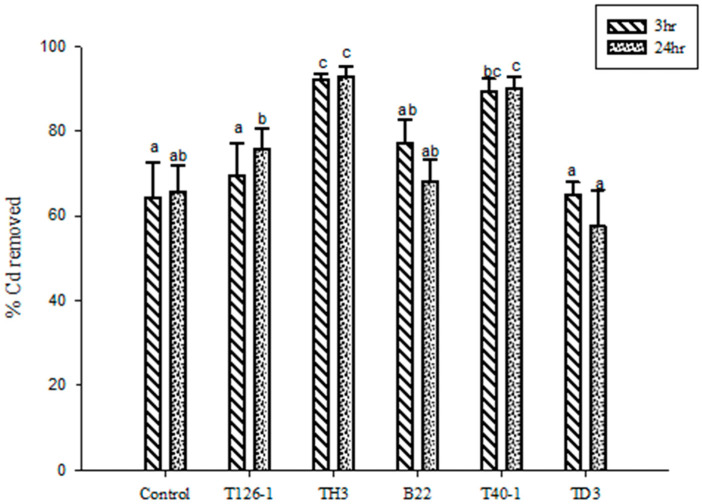
Banana peel fermentation liquid added 50 mg/L cadmium chloride solution for 3 h and 24 h to remove cadmium. ^a,b,c^ Values with different letters indicate a significant difference (*p* < 0.05) using Duncan’s multiple range test.

**Figure 3 foods-12-02632-f003:**
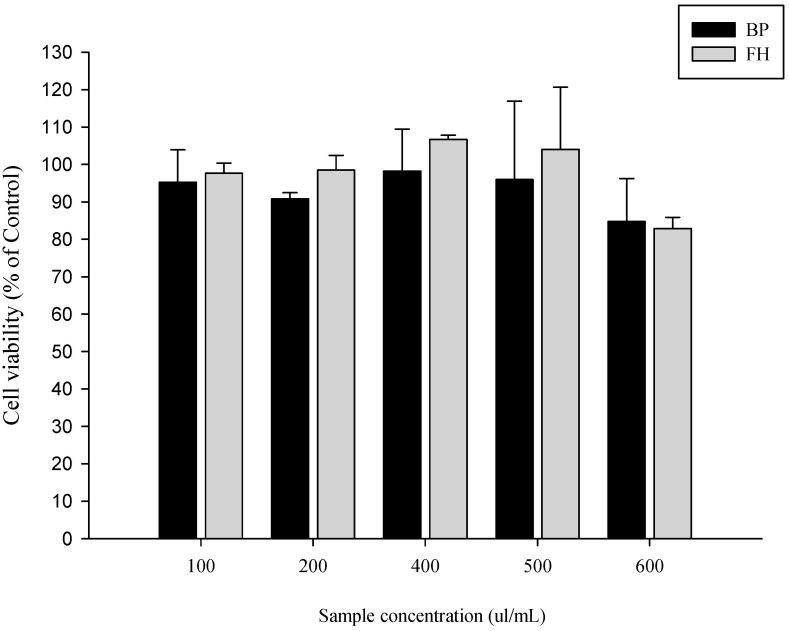
Analysis of the survival rate of Caco-2 human intestinal epithelial cells after 24 h of exposure to fermented products of banana peel lactic acid bacteria. BP: heat treatment of banana peel, FH: heat treatment of banana peel fermented with LAB strain T40.

**Figure 4 foods-12-02632-f004:**
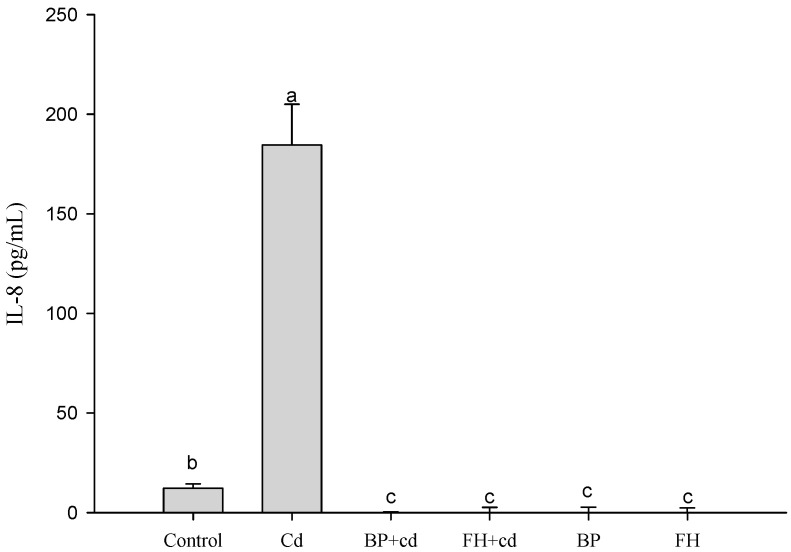
Effects of the fermented products of lactic acid bacteria from banana peel on Caco-2 production of IL-8 in human intestinal epithelial cells induced by cadmium. ^a,b,c^ Values with different letters indicate a significant difference (*p* < 0.05) using Duncan’s multiple range test.

**Figure 5 foods-12-02632-f005:**
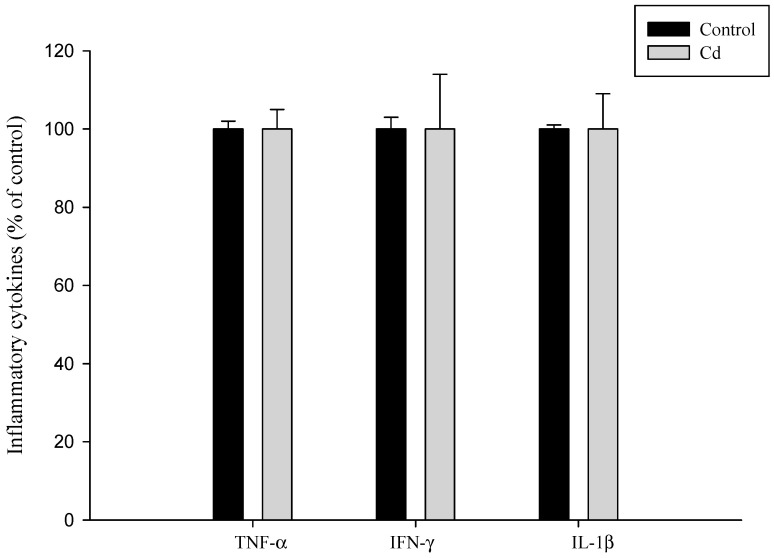
Cadmium induces the cytokine production of human intestinal epithelial cells Caco-2.

**Figure 6 foods-12-02632-f006:**
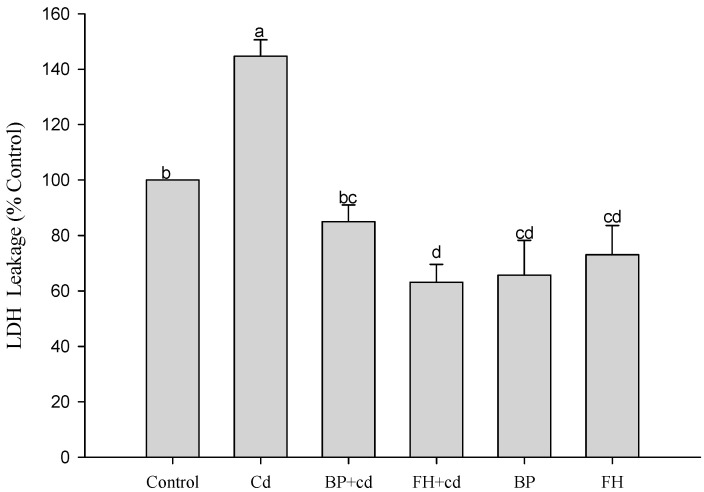
Effects of fermented banana peel by lactic acid bacteria on LDH production of Caco-2 human intestinal epithelial cells induced by cadmium. ^a,b,c,d^ Values with different letters indicate a significant difference (*p* < 0.05) using Duncan’s multiple range test.

**Figure 7 foods-12-02632-f007:**
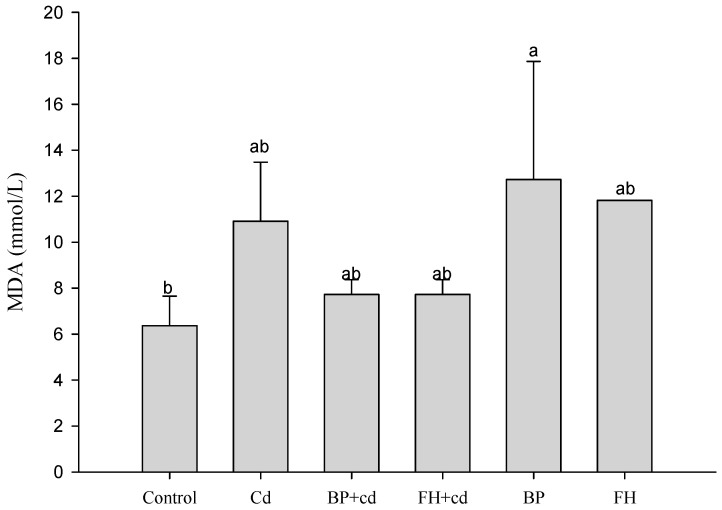
Effects of fermented banana peel by lactic acid bacteria on MDA production of Caco-2 human intestinal epithelial cells induced by cadmium. ^a,b^ Values with different letters indicate a significant difference (*p* < 0.05) using Duncan’s multiple range test.

**Figure 8 foods-12-02632-f008:**
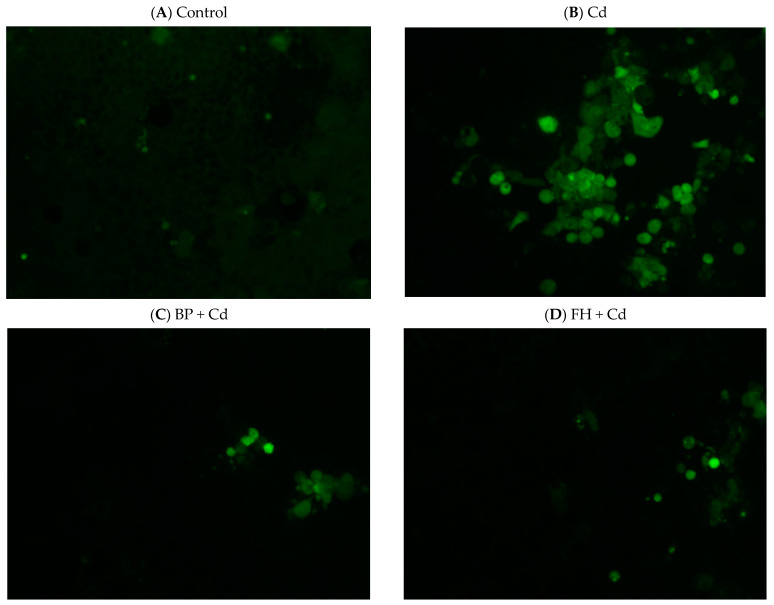
Effects of banana peel fermented by lactic acid bacteria on ROS production of Caco-2 human intestinal epithelial cells induced by cadmium (100×).

**Figure 9 foods-12-02632-f009:**
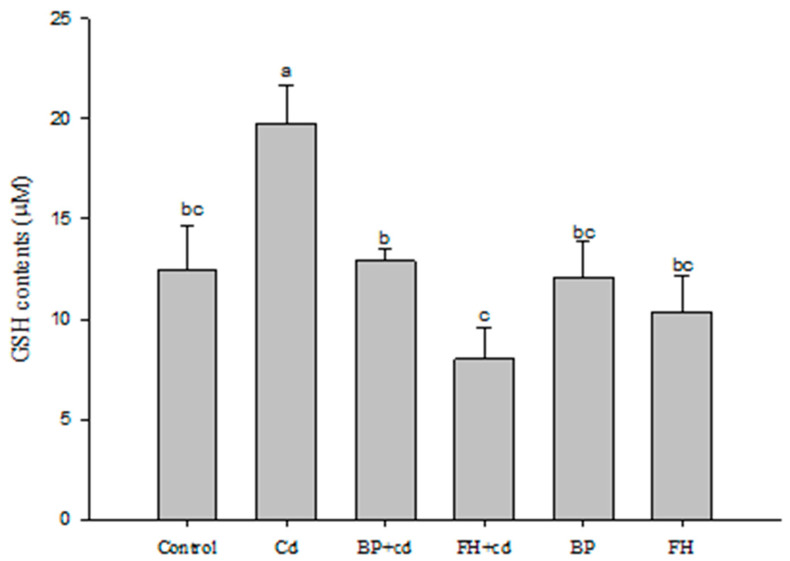
Effects of fermented banana peel by lactic acid bacteria on GSH production of Caco-2 human intestinal epithelial cells induced by cadmium. ^a,b,c^ Values with different letters indicate a significant difference (*p* < 0.05) using Duncan’s multiple range test.

**Figure 10 foods-12-02632-f010:**
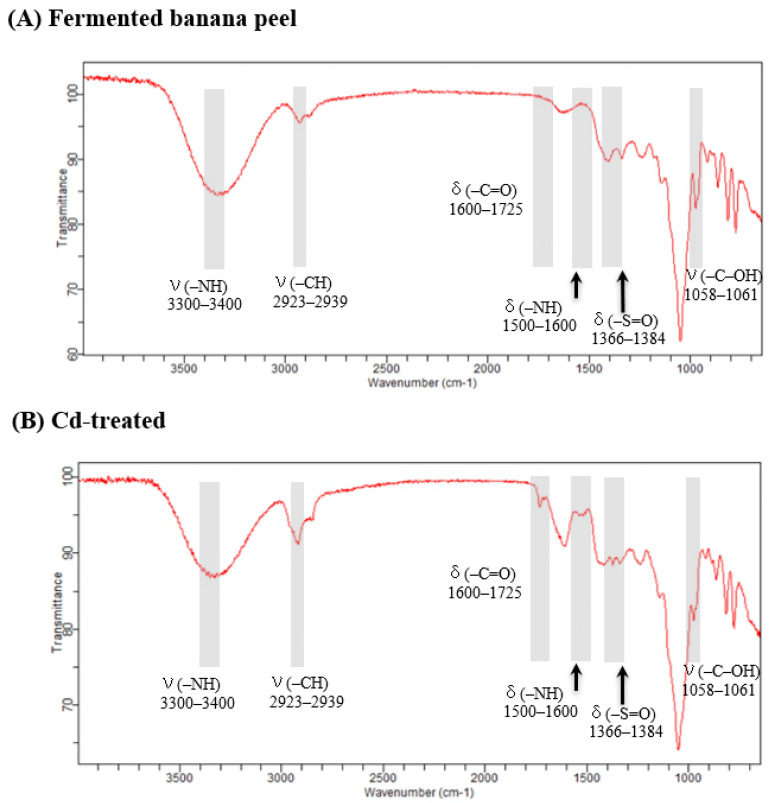
Comparison of the FT-IR spectra of fermented banana peel by lactic acid bacteria T40 and Cd treated in the mid-infrared region (650–4000 cm^−1^). ν: stretching; δ: bending modes.

**Table 1 foods-12-02632-t001:** Lactic acid bacteria for tolerance to cadmium.

Strain	Minimum Inhibitory Concentration at 50% for Cd (mg/L)	
	24 h	48 h
T126-1	500	500
MU1	62.5	62.5
TH3	250	250
T16-1	250	500
B22	250	250
TF-1	250	250
MG2	250	125
T40-1	500	500
TD3	250	125
MK4	125	125
MT3	62.5	62.5

**Table 2 foods-12-02632-t002:** Antioxidant capacity of fermented banana peel by different lactic acid bacteria strains.

Subject	Fermented Supernatant of Banana Peels		
	Control	TH3	T40-1
Scavenging rate of OH (%)	74.85 ± 0.11 ^a^	84.67 ± 0.96 ^b^	80.66 ± 4.27 ^ab^
Scavenging rate of DPPH (%)	86.81 ± 2.67 ^a^	89.95 ± 0.15 ^a^	89.27 ± 2 ^a^
Fe^2+^ chelating ability (%)	28.99 ± 0.59 ^a^	30.25 ± 0.59 ^a^	36.97 ± 0 ^b^
Reducing activity (%)	72.08 ± 6.99 ^a^	75.84 ± 9.61 ^a^	75.11 ± 9.71 ^a^

^a,b^ Values with different letters in the same row indicate a significant difference (*p* < 0.05) using Duncan’s multiple range test.

**Table 3 foods-12-02632-t003:** Lactic acid bacteria ability of acid (pH 3.0) and bile tolerance.

Strain	Acid Tolerance (log CFU/mL)			Bile Salt Tolerance (log CFU/mL)		
	0 h	1.5 h	3 h	0 h	1.5 h	3 h
T40-1	8.7 ± 0.07	8.71 ± 0.05	8.61 ± 0.13	8.63 ± 0.01	8.55 ± 0.04	8.63 ± 0.03
TH3	8.51 ± 0.01	8.4 ± 0.02	8.53 ± 0.04	8.54 ± 0	8.4 ± 0.02	8.57 ± 0.02

**Table 4 foods-12-02632-t004:** The antibiotic susceptibility test for different lactic acid bacteria strains.

Antibiotic	Inhibition Zone (mm)	
	Strain T40-1	Strain TH3
Kanamycin (30 μg)	12 (+)	10 (-)
Ampicillin (10 μg)	30 (+++)	20 (++)
Penicillin G (10 units)	19 (++)	27 (+++)
Streptomycin (10 μg)	10 (-)	10 (-)
Tetracycline (30 μg)	30 (+++)	34 (+++)
Gentamicin (30 μg)	12 (+)	16 (+)
Neomycin (30 μg)	15 (+)	13 (+)
Erythromycin (15 μg)	21 (+)	11 (-)

The inhibition zones ≦11 mm, 12–16 mm, 17–22 mm and ≧23 mm, were classified as strains of no “-”; mild “+”; strong “++”; very strong “+++”inhibition, respectively.

## Data Availability

Data is contained within the article.

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
