# Peer review of "Assessing the Effectiveness of Fermented Banana Peel Extracts for the Biosorption and Removal of Cadmium to Mitigate Inflammation and Oxidative Stress"

_foods, 2023, doi:10.3390/foods12132632_

Round 1

Reviewer 1 Report

In the title remove “heavy metals”

The language needs professional revision, as there are many long sentences that are hard to read and several punctuations mistake “the abstract is a good example for that”.

Introduction:

Contains sufficient information.

But need English editing.

Revise the sentence “ in bioaccumulation issues [2].”

Is this structure is right “LAB fermented with banana peel probiotics”

Did the study contain transmission electron microscopy (TEM) assay as indicated in the aim.

Materials and methods was prepared in a good way with sufficient details.

In statistical section

Specify which tests was used.

Add the right citation of SPSS 20, it must be “IBM SPSS Statistics for Windows, Version 27.0. Armonk, NY: IBM Corp”.

Remove “A P-value less than 0.05 indicated a statistically significant difference.”

Figure 1: revise the caption to be more readable.

Figure 3: what the meaning of BP and FH, use full name in the caption or the legend.

Figure 5. don’t repeat the use of Heavy metal before cadmium, its well known that Cd is heavy meatal. Could you consider along the MS.

Also, the use of abbreviations is incorrect along the MS.

The results and discussion

The results well described but discussion in several parts is missing for instance :

Screening of LAB with cadmium tolerance

Adsorption of cadmium by LAB-fermented banana peel

Determination of probiotic properties

The conclusion is not prepared in right way. The first 17 lines should be removed and can be used to support the discussion. The conclusion must be a take home message based on the results of the current study not conclusion for other several papers.

 Extensive editing of English language required

Author Response

Response to the reviewer’s comments:

  1. In the title remove “heavy metals”

Ans: We have corrected.

  1. The language needs professional revision, as there are many long sentences that are hard to read and several punctuations mistake “the abstract is a good example for that”.

Ans: We have edited in English by MDPI (english-edited-67942) and have English-Editing-Certificate-67942

  1. Introduction: Contains sufficient information. But need English editing. Revise the sentence “ in bioaccumulation issues [2].” Is this structure is right “LAB fermented with banana peel probiotics”. Did the study contain transmission electron microscopy (TEM) assay as indicated in the aim. Materials and methods was prepared in a good way with sufficient details.

Ans: We have corrected. We also have edited in English by MDPI (english-edited-67942) and have English-Editing-Certificate-67942. Please see the revised manuscript. “LAB fermented with banana peel probiotics” changed to “banana peel fermented with LAB”. We have deleted the transmission electron microscopy (TEM) assay

  1. In statistical section: Specify which tests was used. Add the right citation of SPSS 20, it must be “IBM SPSS Statistics for Windows, Version 27.0. Armonk, NY: IBM Corp”. Remove “A P-value less than 0.05 indicated a statistically significant difference.”

Ans: We have corrected. Please see the revised manuscript.

  1. Figure 1: revise the caption to be more readable. Figure 3: what the meaning of BP and FH, use full name in the caption or the legend. Figure 5. don’t repeat the use of Heavy metal before cadmium, its well known that Cd is heavy meatal. Could you consider along the MS. Also, the use of abbreviations is incorrect along the MS.

Ans: We have corrected. Please see the revised manuscript. BP: heat treatment of banana peel, FH: heat treatment of banana peel fermented with LAB strain T40. We have deleted “Heavy metal” before cadmium.

  1. The results and discussion: The results well described but discussion in several parts is missing for instance : Screening of LAB with cadmium tolerance. Adsorption of cadmium by LAB-fermented banana peel. Determination of probiotic properties

 Ans: We have added the discussion for “Screening of LAB with cadmium tolerance. Adsorption of cadmium by LAB-fermented banana peel. Determination of probiotic properties”. Please see the revised manuscript.

  1. The conclusion is not prepared in right way. The first 17 lines should be removed and can be used to support the discussion. The conclusion must be a take home message based on the results of the current study not conclusion for other several papers.

Ans: The first 17 lines have been removed.

  1. Comments on the Quality of English Language:Extensive editing of English language required

Ans: We have edited in English by MDPI (english-edited-67942) and have English-Editing-Certificate-67942.

Reviewer 2 Report

Due to the absence of line numbers, the information on the suggestions was pointed out in the copy of the manuscript attached to this revision.

The manuscript repeats Tables and Figures at the end of the manuscript. Since they were previously included in the text, they should be removed.

See comments in the attached manuscript.

Author Response

Response to the reviewer’s comments:

1.Due to the absence of line numbers, the information on the suggestions was pointed out in the copy of the manuscript attached to this revision.

The manuscript repeats Tables and Figures at the end of the manuscript. Since they were previously included in the text, they should be removed.

Ans: We have made the necessary corrections to address the issues raised by the review committee. Please refer to the attached file. Duplicate Figures and Tables have also been removed.

Round 2

Reviewer 1 Report

The manuscript has been improved and could be accepted in the current form.